# Nutritional self-care practices and skills of patients with diabetes mellitus: A study at a tertiary hospital in Ghana

**Kwabena Opoku-Addai[1,2], Kwadwo Ameyaw Korsah[1], Gwendolyn Patience Mensah**[1]*

**1** School of Nursing and Midwifery, College of Health Sciences, University of Ghana, Legon, Accra, Ghana, **2** Faculty of Health and Medical Sciences, Department of Nursing and Midwifery, Presbyterian University College, Asante Akyem, Agogo, Ghana

* gpmensah@ug.edu.gh

## Abstract

### Introduction

Nutritional management decreases and/or prevents the complications and deaths associated with diabetes mellitus. However, the majority of patients living with diabetes do not engage in optimal nutritional management of diabetes because they see it as the most difficult aspect of managing the condition. This study aimed to explore and describe the practices and skills on nutritional management of diabetes mellitus among patients living with diabetes attending a Ghanaian hospital.

### Materials and methods

This study employed an exploratory, descriptive qualitative research design. Fifteen participants were recruited using purposive sampling, and interviewed with a semi-structured interview guide. Content analysis was performed on the data gathered, following which three main themes emerged.

### Results

More than two-thirds of the participants of this study had adequate meal planning skills, ate the right quantity of foods, engaged in healthy eating habits, and consumed healthy sources of carbohydrates, fats and protein when eating. However, more than half of the participants had insufficient knowledge and skills in the reading and usage of food labels.

### Conclusions

The participants of this study largely engaged in optimal nutritional management of diabetes due to their healthy dietary practices and preferences. It is recommended that health care professionals in Ghana find practical and robust ways to factor the reading and usage of food labels into the care and management of patients with diabetes.

**Data Availability Statement:** All relevant data are within the paper.

**Funding:** The authors received no specific funding for this work.

**Competing interests:** The authors have declared that no competing interests exist.

**Abbreviations:** NDMRC, National Diabetes Management and Research Centre.

## Introduction

Diabetes mellitus is currently the world's most threatening epidemic, with low-income and middle-income countries particularly experiencing this effect [1]. Africa recorded a diabetes prevalence of 7.1% in 2014, which is a 129% increase in the prevalence of the condition since 1980 [2]. That notwithstanding, the high prevalence of diabetes mellitus in sub-Saharan Africa is likely to be doubled by 2035 [3].

Diabetes can result in a variety of complications such as lower limb amputation, stroke, heart attack, blindness and kidney diseases, when it is untreated, uncontrolled or poorly managed [4]. In sub-Saharan Africa, the ascendency of diabetes and related complications are currently a threat to the health systems. The condition and its complications have the potential to overwhelm health systems, dwindle the expenditure of households and reverse some of the achievements that sub-Saharan Africa has seen in many other health outcomes in recent years [2].

In 2012, diabetes accounted for 1.5 million deaths globally. In the same year, 2.2 million deaths were attributed to cardiovascular and other diseases stemming from high blood glucose levels [4]. The number of mortality due to diabetes increased significantly in 2017. This is because four million people died as a result of the condition, representing one death in every eight seconds. This number of mortality caused by diabetes surpassed the combined number of mortality caused by HIV/AIDS, malaria and tuberculosis in 2015 [5]. In Ghana, diabetes mellitus was the ninth leading cause of death among patients admitted in 2016 [6]. Provision of supportive environments for living healthy lifestyles and the treatment of diabetes could have prevented the majority of these deaths and complications [4]. The morbidity and mortality rate associated with diabetes can therefore be significantly reduced by improving the control of diabetes through education, lifestyle modification, and effective treatment [7]. Research studies have revealed that nutritional management is pivotal to lifestyle modification, diabetes education, treatment, care and management [8,9].

Nutritional management decreases and/or prevents the complications and deaths associated with diabetes mellitus, and also decreases the health expenditure of patients with diabetes [9,10]. Best possible glycaemic control and improved quality of life for patients with diabetes are also dependent on optimal nutritional management [8,11,12]. Nutritional management has therefore been touted as the cornerstone of successful management of diabetes mellitus [13–16]. Research studies have also posited that foundational to a successful nutritional management of diabetes mellitus is the need for patients with diabetes to possess the required skills and abilities to engage in healthy eating [17,18].

However, the majority of patients living with diabetes do not engage in optimal nutritional management of diabetes because they see it as the most difficult aspect of controlling the condition [13]. Optimal nutritional management of diabetes mellitus involves engaging in healthy dietary intake and practices, reading and using food labels, and planning appropriate meals [19].

To the best of the researchers' knowledge, there is scanty information on the nutritional skills and practices of patients living with diabetes mellitus in Ghana. Thus, this study sought to explore and describe the practices and skills on nutritional management of diabetes mellitus among patients with the condition who were attending clinic at a tertiary hospital in Ghana. The findings of this study will enable health education programmes to be designed to improve the practices and skills of patients with diabetes on the nutritional management of their condition.

## Materials and methods

### Study area and research design

The study was conducted at a National Diabetes Management and Research Centre (NDMRC) in Accra, Ghana. The NDMRC is a well-resourced centre for diabetes management, research and education, which is managed by highly qualified medical and paramedical staff. The centre provides all the pertinent services and support key to proper diabetes care and management. It has a diabetic clinic which serves as the outpatient department and provides medical care to patients with diabetes.

An exploratory, descriptive qualitative research design was used for this study.

### Sample size and sampling technique

Purposive sampling technique was used to recruit participants for the study. The target population was all patients with diabetes attending clinic at the NDMRC. The inclusion criteria comprised all patients with type 1 or type 2 diabetes mellitus who were attending clinic at the NDMRC, were eighteen years and above, and had been diagnosed of diabetes mellitus for at least six months. The sample size of a qualitative research depends on the saturation of data, that is, the giving of similar responses by successive participants, and the emergence of no new theme or subtheme [20]. This study comprised of fifteen participants. This is because saturation of data was reached by the fifteenth participant.

### Data collection

Permission was sought from the tertiary hospital's Institutional Review Board (ethics number: KBTH-IRB /000130/2018) and the head of the NDMRC. Patients with diabetes attending clinic at the NDMRC were then informed about the study and requirements for participation. Guided by the inclusion criteria, the researchers recruited participants who willingly agreed to be part of the study for face-to-face interviews.

Before an interview was conducted, each participant was given a consent form to read, and invited to ask questions and seek clarifications based on the content of the consent form and the study. Each participant then voluntarily signed or thumb printed the consent form after he or she had sought clarification for all doubts and was satisfied with the information provided. For participants who could not read the consent form, the information was translated to them in the local Twi language which they understood.

Data was collected from the participants using a semi-structured interview guide (S1 file). The lead author who has received education and training on conducting qualitative interviews conducted the interviews. The interviews were conducted with the aid of an interview guide, which was initially piloted at a University Hospital in Ghana on three patients with diabetes. The interview guide comprised of open-ended questions on the participants' daily dietary intake, practices and habits regarding the nutritional management of diabetes mellitus. They were also questioned on their skills and current practices on food label reading and meal planning. Further probes were made based on participants' responses.

The semi-structured interviews, which had a duration of forty to sixty minutes, were conducted in either English or the local Twi language, depending on the dialect a participant was comfortable with. With participants' permission, each interview was audio recorded. To ensure anonymity, the participants were given pseudonyms.

### Data analysis

Braun and Clarke's [21] method of data analysis was used to analyze the data gathered. The audio recorded interviews conducted in English were first transcribed verbatim. Those that

were conducted in the local Twi language were also translated and transcribed verbatim into English by the researchers with the help of a translator to avoid distorting the data. The researchers then read and re-read the data set of fifteen transcripts in English, and noted significant ideas that emerged from the data. These ideas were coded to generate initial codes and matched with relevant extracts from the data. The researchers collated the initial codes into potential themes. The potential themes with their extracts were then reviewed to generate three main themes. A final analysis of the themes was done to come up with a research report which provided accurate answers to the research questions.

## Results

All fifteen participants recruited and interviewed were Ghanaians. The number of years they had been diagnosed with diabetes mellitus ranged from 2 years to 30 years, whiles their ages ranged from 42 years to 86 years. Out of the fifteen participants, seven were males and eight were females.

Using content analysis, three main themes emerged from the data collected. The themes included: dietary practices in the morning, afternoon, and evening; reading of food labels; and meal planning. Below is a presentation of the themes with direct quotes from the participants.

### Dietary practices in the morning, afternoon, and evening

The participants mentioned the number of times they generally eat in a day, the foods they usually eat, the quantity of foods they eat, and their practices on fruits consumption. They also indicated their current practices on the intake of soft drinks and alcoholic drinks. The participants revealed that they avoid starving themselves by typically eating twice or three times a day. More than two-thirds of the participants said that they often eat three times a day. One of them narrated:

*I often eat three times, that's what the dietitian told me.* (Ama).

One-fourth of the participants however specified they usually eat two times a day. One of them recounted:

*I eat twice in a day most of the time, because I don't really like food.* (Yaa).

All participants revealed they habitually eat breakfast. Almost all of them said they usually eat corn porridge, millet porridge, oats or wheat for their breakfast, due to the high fibre content. They often take these foods with whole wheat bread and sometimes add vegetables to the bread. Additionally, some participants indicated they sometimes add powdered milk or a cholesterol-free milk and non-nutritive sweeteners to their foods. One such participant reported:

*For breakfast, I usually take corn porridge or millet porridge, because they* [dietitians, nurses and doctors] *told us to eat foods rich in fibre. I always take the porridge with wheat bread and usually add dandelion, cabbage and carrots to the bread. I don't often add sugar and milk to my food, but sometimes I add the diabetes sugar and Carnation milk or powdered milk to my porridge. I don't take white sugar. The dietitian, nurses and doctors said it is not good for us, so I only take the diabetes sugar.* (Adwoa).

Participants also emphasized that they do not consume large quantities of the high fibre foods they take as breakfast. One participant remarked:

*They* [dietitians, nurses and doctors] *have advised us concerning the amount to take. They said two slices of bread is okay. Two thin slices. We don't eat plenty. They told us about the amount of porridge, oats and wheat to take. They told us to take only two or three ladles, that's all.* (Kofi).

More than half of the participants said they usually have tea with whole wheat bread as their breakfast. They often add powdered milk or a cholesterol-free milk to the tea, and sometimes add vegetables or egg to the bread. However, whereas some of them sometimes add a non-nutritive sweetener to their tea, a few of them usually add white sugar to their tea. One participant stated:

*What I usually eat in the morning is tea with wheat bread. I don't add sugar to it. I only add a tablespoon of powdered milk to it. I sometimes add egg to the bread. But in order not to increase my cholesterol level, I boil the egg, I don't fry it, and I place the boiled egg in the bread.* (Kwabena).

On the quantity of tea and bread they usually take, participants said they take one cup of tea and a small quantity of bread. One of them recounted:

*I take just a cup of tea. The cup I use is the common tea mug. For bread, I take just three slices of bread.* (Kwadwo).

One-third of the participants mentioned that they sometimes take "kenkey" (a local Ghanaian food prepared with corn) with fish and stew or vegetables during breakfast. One participant remarked:

*I sometimes take kenkey with fish in the morning. I take it with tomato stew, kontomire* [spinach] *stew, or cabbage stew. Sometimes, I also take it with ground pepper, sliced tomatoes, and sliced onions.* (Abena).

In the narratives provided by participants, they reported eating a ball of kenkey, the size of which is equal to their closed fist. Abena commented:

*I just go in for kenkey. The size is like my closed fist.* (Abena).

One-fourth of the participants also mentioned boiled plantain with vegetable stew and fish as one of the foods they take as breakfast. From their accounts, they usually take two to three fingers of plantain. Akua recounted:

*By 9am or 10am, I will go in for boiled plantain. I grind my kontomire, add a small amount of oil to it and take it with the plantain. I take it with fish. I take just three fingers of plantain.* (Akua).

Rice with stew, fish, and vegetables was mentioned by one-third of the participants as food they take as breakfast. Participants provided information on the quantities they eat. One of them remarked:

*Sometimes I eat white rice with kontomire stew and fish or brown rice with vegetables, tomato stew and fish in the morning. For the quantity, I take three ladles of rice because they*

[dietitians, nurses and doctors] *said we should take two to four ladles of rice, so that is what I do.* (Akosua).

The narratives of the participants indicated that they usually take their breakfast from 6am to 10am everyday. One of them commented:

*I take my breakfast anytime from 9am to 10am.* (Dede).

In addition to the discussions on breakfast, the participants mentioned the foods they usually take as lunch, as well as the quantity they take. More than half of the participants said they regularly eat "fufu" (a Ghanaian food prepared with only plantain or with cassava and plantain) with soup and fish as their lunch. They usually take their fufu with groundnut soup, palm nut soup or light vegetable soup. The following participant's account is similar to others in this group:

*I often eat fufu in the afternoon. I take it with light vegetable soup, palm nut soup or groundnut soup. I usually take it with fish, but occasionally, I take it with chicken.* (Kwabena).

The participants mentioned that they consume large quantities of soup, but the size of the fufu they usually eat is equal to either one or two of their closed fists. One participant remarked:

*For the fufu, I have my limit, I don't take much. The size of the fufu I take is like my fist. It's the soup that I take a lot.* (Afia).

Half of the participants reported that they usually eat boiled plantain with vegetable stew and fish for lunch. A few sometimes eat it with egg instead of fish, while some occasionally have boiled yam instead of boiled plantain. Yaw commented:

*I usually take boiled plantain with kontomire stew, or sometimes cabbage stew or garden eggs stew. I usually take it with fish. But sometimes I go in for egg. Also, sometimes, I take boiled yam instead of boiled plantain.* (Yaw).

On the quantity of boiled plantain and boiled yam the participants take, they reported that they often consume two to three fingers of plantain, and four to five slices of yam. One of them recounted:

*I take two fingers if the size of the plantain is big, but I take three if the size is small. For yam, I sometimes take four slices, other times too I take five slices of yam.* (Akosua).

Almost half of the participants indicated that they usually have rice and stew with fish as lunch. They often take the rice with tomato stew or kontomire stew, and sometimes add chicken or egg but rarely add red meat to the food. One participant said:

*I take rice with tomato stew or kontomire stew. I usually take fish. Sometimes too I take it with chicken or egg.* (Yaa).

Participants concurred that they usually consume two to four ladles of rice. One stated:

*I don't take much of the rice. I take about three ladles.* (Abena).

One-fifth of the participants mentioned kenkey with fish and ground pepper or okra stew as one of the meals they sometimes have for lunch. One stated:

*With the kenkey, I take it with fish. If I have okra stew, I use that. If I don't have, I prepare pepper. I buy one ball of kenkey, which is like my closed fist.* (Ama).

One-fifth of the participants also said that they sometimes eat "banku" (a local Ghanaian food prepared with corn and cassava dough) or "rice-ball" (a Ghanaian food prepared with rice) with soup and fish or chicken for lunch. They usually consume one or two fist-sized balls of banku or rice-ball. One of them remarked:

*I usually eat banku or rice ball in the afternoon. I take it with light vegetable soup, groundnut soup, or palm nut soup. I take them with dry fish and chicken, and on some few occasions I take them with red meat. I only take a small quantity of it—it is like my fist.* (Yaa).

Participants indicated that they often have their lunch from 12pm to 3pm, as one commented:

*I eat by 1pm in the afternoon. So I take my lunch anytime from 12pm to 1pm.* (Abena).

On their supper intake, the participants said they usually eat early in the evening, and when they do, they take in heavy foods. However, most of them concurred that when they become hungry again later in the evening, they eat light foods due to the education they have received to avoid starving themselves and eating heavy foods at night. The types of heavy foods the participants often have for supper are not different from the ones they usually have as lunch. Similarly, the types of light foods they consume when they feel hungry later in the night are not different from the light foods they usually eat in the morning.

Two-thirds of the participants said they usually eat a fist-sized ball of fufu with light vegetable soup, groundnut soup or palm nut soup and fish as their supper. Sometimes, a few of them add red meat to their fufu.

Half of the participants reported that they usually consume banku, which size is like their closed fist, with fish and vegetable stew as supper. One commented:

*I take banku with okra stew and fish. Sometimes, I also take it with pepper, sliced tomatoes, sliced onion, and fish—usually tilapia. The size of the banku I take is like my closed fist. When I am very hungry, I add half the size of my closed fist to it. But I take more of the okra stew, and if I'm taking it with pepper, I take more of the sliced tomatoes and onion.* (Akosua).

Almost half of the participants mentioned boiled plantain with vegetable stew and fish as food they eat in the evening. They consume only two or three fingers of plantain, as one participant remarked:

*I usually take boiled plantain with dandelion stew, kontomire stew or garden eggs stew, and fish. As for plantain, I take just two fingers.* (Ama).

One-fifth of the participants said that they sometimes eat boiled yam with stew and fish, and take only four or five slices of yam. One participant recounted:

*I sometimes take yam. I take about five slices. I take it with kontomire stew, cabbage stew or garden eggs stew and fish. I take four to five slices of the yam because that is what they* [dietitians, nurses and doctors] *told me.* (Yaw).

More than half of the participants indicated they usually eat either one ball of kenkey with ground pepper and fish or rice with stew, vegetables, and fish in the evening. They stated they do not eat much of the kenkey or rice. One of them said:

*In the evening I take rice. For the rice, I add vegetables and fish, then stew. I eat two or three ladles of rice. I sometimes also eat kenkey and pepper with fried fish in the evening. I eat just one ball of kenkey.* (Kwaku).

On the time they usually have their supper, almost all the participants said they generally eat their supper from 4pm to 7pm. One of them commented:

*I eat in the evening. I usually eat around 6:30pm till 7pm.* (Yaa).

However, a 63-year-old man who has been living with diabetes for 25 years said he usually has his supper between 6pm and 10pm due to the nature of his work:

*In the evening I eat anytime from 6pm to 10pm. Because of the nature of the job I'm doing, I sometimes eat late. I don't have time for my food because of the nature of my job.* (Kwaku).

More than half of the participants also mentioned that they occasionally eat light foods when they become hungry later in the evening, after taking their supper. One recounted:

*Once a while, I feel hungry when I'm about sleeping. If I am hungry at that time, I eat. Around that time, I don't eat any heavy food, I just take in something light like oats, porridge, tea, banana, or biscuit. And I don't eat much.* (Kwame).

On their practices concerning the consumption of fruits, all participants indicated that they take fruits regularly. Comments included:

*I take a fruit everyday. Every blessed day I take a fruit.* (Kwabena).

*I take fruits three to five times a week.* (Yaw).

Half of the participants mentioned that they often eat fruits in the evening, while one-third of the participants shared that they usually have fruits in the afternoon. One of them recounted:

*For me, it is in the afternoon that I often take my fruits. I take some before I take my lunch. Even after lunch I take some too.* (Abena).

One-fifth of the participants also indicated they often consume fruits in the morning. One person remarked:

*I usually try to eat my fruits before taking my breakfast, or sometimes together with my breakfast. So usually, breakfast will be set, and the fruits will be set alongside too.* (Kofi).

The participants cited orange, banana, pawpaw, mango, coconut, pineapple, apple and pear as the fruits they often consume. Statements in this respect included:

*I usually take a lot of pawpaw, banana and orange. I take mango too.* (Yaa).

*I usually take apple or coconut. Sometimes, I take pineapple. Even pear, I take that one sometimes.* (Dede).

The participants also talked about their practices concerning the consumption of soft drinks and alcoholic drinks. More than two-thirds of the participants indicated that they currently take soft drinks in moderation and occasionally, dilute Coca-Cola and Fanta with water before consuming it, and only take half a glass of non-alcoholic malt drink. A few of the participants added that they often consume Coca-Cola when their sugar level is very low. Responses on this topic included:

*I don't take soft drinks that often. I only take them once in a while. If I'm taking a drink like Coca-Cola or Fanta, I pour half of it away and mix the remainder with one sachet of water before taking it. For Malt, I only take half the amount, that's all. But I don't mix the half that I take with water.* (Kwame).

*I take Coca-Cola when my sugar level goes down. The doctors and nurses have told us to take Coca-Cola when our sugar level drops, so that's what I do. But apart from that, I take Coca-Cola, Malt, Fanta or Soda water occasionally.* (Akosua).

In contrast, one-fifth of the participants echoed they do not consume soft drinks. One of them said:

*I don't even like soft drinks because they will give me health problems, so I avoid them.* (Adwoa).

More than two-thirds of the participants also indicated that they do not take alcohol. Two of them stated:

*As I told you, I don't drink. I wasn't drinking even before diagnosis, and up till now I don't drink.* (Kwaku).

*Before diagnosis I was taking alcohol, but ever since I was diagnosed, I have stopped.* (Akwasi).

In contrast, a few of the male participants admitted that they currently consume alcohol occasionally and in moderation. One of them remarked:

*Occasionally, I take beer. Now, I don't take a bottle of beer, I often take a glass of beer. I've now limited how I take it. It's now different from how I was taking alcohol before I got diabetes. It is on rare occasions that I even take a bottle of beer now. But even with that, I don't take more than a bottle of beer now.* (Yaw).

## Reading of food labels

The participants of this study shared their views on the education received on the usage of food labels, and on their current practices of reading and using food labels in the nutritional management of their condition. Half of the participants recalled that they were educated on

the use of food labels by doctors, nurses and dietitians. One-third of the participants indicated that they only check the expiry date of processed foods because that is what they were told to do, and that they rarely consume processed foods. One of them remarked:

*They* [dietitians, nurses and doctors] *educated us to check the expiry date of processed foods. For me, I don't like canned foods, so I hardly buy them. If I buy those products, expiry date is what I check. I often check the expiry date of such products.* (Afia).

Less than one-fifth of the participants said that they were educated to check both the expiry and manufacturing dates of processed foods. One of them recounted:

*They* [dietitians, nurses and doctors] *educated us on the food labels. They said the manufacturing date and expiry date are very important, so we should always check them before we buy canned foods or processed foods. I always make sure I see the manufacturing and expiry dates on the labels of food products before I buy them. If I don't see the dates, I won't buy the food product.* (Kwaku).

One participant indicated that he was educated to check the sugar content and expiry date of processed foods before consuming them, and that is what he does. He explained:

*Personally, I don't like processed foods. But if I'm buying them, what I check on those things is the expiry date. If it has expired, I won't buy it. I know that all those foods contain sugar, so I check the sugar content too. I often check the expiry date and the sugar content of those foods, because that is what they* [dietitians, nurses and doctors] *told me to do.* (Kwabena).

In contrast, a few of the participants said that they were not given education on the reading and usage of food labels by their health care providers, but received that education from watching or listening to health programmes on the television and radio. They indicated that what they check on food labels is the expiry date of the products. One of them commented:

*The health personnel didn't teach us about food labels. I heard about food labels on the television when I was watching a health program. They said that canned foods and bottled foods are not good for diabetics so we need to avoid them. But if we have to buy them, we need to check their expiry dates. So when I have to buy such products, I only check the expiry date.* (Adwoa).

One-third of the participants confirmed they had not received education on the use of food labels from their health providers or from any other source. One-fourth of the participants also said they do not check anything when they buy processed foods. One of them stated:

*The health workers haven't taught us anything about food labels. They didn't tell us that, so I don't know anything about food labels. I don't know the meaning of the things there, so I don't check for anything before I buy those products. But for me, I don't like the canned foods and bottled foods, so I hardly buy them.* (Akua).

It was the habit of one participant to check the expiry date of processed foods even though she indicated that she was not educated on the use of food labels. She remarked:

*I wasn't given any education on food labels by the health personnel, but I always check the expiry date of those products. I also check if they have written 'sugar free' on it. Those are the two things I often check for.* (Abena).

## Meal planning

Study participants shared how they plan their meals and the factors they consider when planning their meals. More than two-thirds of them mentioned that in opting for a particular food, they consider whether they have been asked by their doctors, nurses and dietitians to eat that food or to avoid it, whether the food will increase their blood glucose level or not, and whether the food is healthy for them or detrimental to their health. One participant explained:

*For me, I only check if the food will not raise my sugar level. So if it is something I have been told by the nurses, doctors and dietitian to avoid, I will not go near it. But if it is something that I have been asked to eat, that one I know it will not give me problems so I will eat it.* (Akosua).

However, one-fifth of the participants said their desire for a particular food is the only thing they consider before opting for that food. One of them said:

*What I have appetite for is what I consider. So, it is if I have the appetite for this food or that particular food. That's all.* (Yaa).

## Discussion

Regular breakfast consumption, particularly whole grains and cereal diets, are positively associated with healthy glycaemic levels, healthy body weight, and decreased risk of cardiovascular conditions [22,23]. This study found that most of the participants usually ate three times daily, that is, breakfast, lunch and supper. A few of the respondents however reported that they often ate two times daily.

All the participants of this study were regular breakfast consumers. The findings of the study highlighted that the foods the participants usually ate at breakfast generally consisted of high amounts of whole grains, cereals, vegetables, milk, fish and non-nutritive sweeteners, as well as low to moderate amounts of oil, legumes, fats, cholesterol, meat and sugar. This suggests that the participants consume healthy sources of carbohydrates, protein and fat during breakfast. These findings partly corroborate the findings of a study conducted among patients with diabetes in Mauritius which showed that the breakfast of the participants usually contained high amounts of vegetables, fish, and milk [24]. The findings also concur with the findings of a study conducted in the United States of America which found that the participants usually consumed high amounts of vegetables, and low amounts of sugar during breakfast [25].

Patients with diabetes need to be regular lunch consumers in order to achieve positive health outcomes such as the maintenance of a healthy body weight and healthy blood glucose levels [26]. Most of the participants of this study reported being regular lunch consumers. They usually consumed high amounts of staple foods, vegetables and fish; low amounts of red meat; and moderate amounts of eggs, chicken, cereals, legumes, oil and whole grains during lunch. Hence, the participants generally consume healthy proteins, carbohydrates and fats during lunch. These findings resonate with the findings of a study conducted in China by Li et al.

[27] which showed that people who regularly have lunch often consume increased amounts of vegetables. The findings also partly align with the findings of a study by Griffith, Wooley and Allen [28] which found that the participants of that study usually consumed moderate amounts of whole grains, poultry, dairy products, and vegetables during lunch.

Irregular supper intake and late night eating among patients with diabetes lead to poor glycaemic control [29]. The findings of this study indicated that the participants were regular supper consumers who often ate their supper early, and avoided late eating and night-time starvation. This is partly in tandem with the findings of studies by Sandhu and Tang [30] and Sakai et al. [29] which found that the majority of participants were regular supper consumers who avoided late night eating by usually having their supper before 8pm. Generally, the participants of the current study eat healthily during supper. Foods they reported taking as supper included high amounts of vegetables, whole grains, fish, roots and tubers; moderate amounts of legumes, cereals, fats and oils; and low or rare amounts of red meat, poultry and dairy products. These findings are in contrast with the findings of a study conducted in Japan by Sato-Mito et al. [31] which showed that during supper, the participants often consumed foods containing high amounts of red meat, refined grains, fats, oil and sugar, as well as low amounts of vegetables, whole grains, cereals, fruits and milk. The Japanese study also indicated that the participants often engaged in late night eating, as well as the consumption of fast foods, sweets, sugary drinks and refined grains when taking supper. The reason for the contrasting findings may be due to the choice of foods the participants of both studies often consume during supper. The healthy sources of carbohydrates and protein contained in the diets of the participants of the present study as compared to the unhealthy carbohydrate and protein sources in the diets of the participants of the Japanese study may have accounted for the different findings.

The participants of this study kept in check the quantity of foods they ate and avoided overeating to ensure optimal glycaemic control. They consumed two to four ladles of porridge, oats, wheat, soup and rice, as well as took a specific number of slices of bread. They also consumed specified portions of the local Ghanaian foods of kenkey, banku, fufu and rice-ball—which sizes were as their closed fists, ate specific fingers of plantain, specific slices of yam, and only took one cup of tea. Hence, the participants were clear on the amount of foods to eat and consistent on the quantity of foods they consumed during breakfast, lunch and supper. These findings are contrary to a study carried out by Uchenna et al. [32] among patients with diabetes in Nigeria which showed that the participants were unable to estimate the recommended quantity of foods to eat, and were eating inappropriate sizes of foods despite receiving education on the quantity of foods to eat from their health professionals. The Nigerian study further indicated that the participants had a poor understanding of the actual quantity of foods to eat because they were only told to generally cut down on the amount of foods they eat without receiving specific instructions on the actual amount of foods to eat. Therefore, the lack of clear instructions from the health providers of the participants of the Nigerian study and the reportedly poor health education ability and skills of the health personnel in Nigeria may have resulted in the contrasting findings.

The participants of this study consumed appreciable amounts of several fruits daily. It appeared they often took fruits as their snack, consumed high amounts of fruits in the evening, moderate amounts in the afternoon, and low amounts in the morning. The common fruits the participants eat include orange, banana, pawpaw, mango, coconut, pineapple, apple and pear. This is similar to the findings of another study conducted among patients with diabetes in another part of Ghana [33].

Patients with diabetes need to limit their intake of sweets and soft drinks to avoid increasing their lipid levels and blood glucose levels [34]. The majority of the participants of the present study reported that they currently consumed soft drinks occasionally and in moderation to

keep their blood glucose levels in a healthy range. However, a few of the participants reported that they had stopped consuming soft drinks to avoid raising their glucose levels. These findings are in line with the findings of a study carried out in Brazil among patients with diabetes which found that the majority of participants consumed soft drinks occasionally and in moderation to avoid increasing their blood glucose levels [34].

Almost all the participants of the current study reported that they do not drink alcohol. However, two male participants reported drinking alcohol occasionally and in moderation. These findings are congruent with the findings of studies carried out by Yoshimura et al. [35] and Ewers et al. [36] which respectively showed that men with diabetes consume alcohol more than women with diabetes; and patients with diabetes often consume only moderate amounts of alcohol because of their condition. The findings of this study also resonate with the findings of a study conducted among patients with diabetes by Jakobsen et al. [37] in which the majority of the participants reported that they had not taken alcohol within the past year. Various research studies have posited that patients with diabetes should consume moderate amounts of alcohol, while others have emphasized that patients with diabetes should totally refrain from consuming alcohol because of the risks and negative effects of moderate alcohol consumption on health outcomes [38]. Though further research is needed in this area for better understanding, it currently appears safe for patients with diabetes to avoid consuming alcohol, even in moderation, because the health benefits associated with abstinence tend to outweigh the benefits associated with moderate alcohol consumption in the management of diabetes [38].

Proper understanding and usage of food labels involve the ability of patients with diabetes to consider the amount of sugar, fibre, energy, fat and salt on packaged foods, as well as the expiry dates of these products, before buying and consuming them [39]. The findings of this study indicated that the participants restricted their food label reading to checking expiry dates, the sugar content and manufacturing dates of processed foods. This is consonant with the findings of a study carried out by Washi [40].

The findings of the present study also appear to support the assertion that, there is a positive association between nutritional knowledge and food label usage since all the participants who received education on food label reading practiced what they were told. The findings however suggest a deficient knowledge and skills on food label reading among the participants and their health providers. It is recommended that workshops should be organized for the health providers at the NDMRC on the reading and usage of food labels, and how they can effectively integrate this information into the care and management of patients with diabetes. Furthermore, it is recommended that the Government of Ghana formulate a policy for companies that manufacture processed foods in the country to use local languages on the food labels of their products in addition to English, to enhance the usage of food labels by people who cannot read English. Finally, the findings suggest that there is a need for further research to explore the knowledge and practices of health professionals in Ghana on the education of patients with diabetes, relating to the reading and usage of food labels.

The narratives of the participants in this study illustrate that the successful management of the nutritional aspects of living with diabetes requires meal planning. A meal plan is a guideline given by health professionals to patients with diabetes to enable them engage in healthy eating. Meal planning is geared towards enabling patients with diabetes to eat a variety of foods, consume quality and healthy carbohydrates, fats and protein, and also keep the quantity of foods they eat in check without necessarily measuring foods, weighing foods, and estimating the percentages or total amount of carbohydrates, fats and protein that they eat daily [41,42]. Additionally, meal planning also involves the ability of patients with diabetes to eat regularly, avoid skipping meals, and avoid excessive consumption of snacks in between meals [19]. Various research works have therefore touted the significance of employing the Mediterranean diet

in the planning of meals for patients with diabetes since it enables them to consume healthy food sources and eat the right quantity of foods. The Mediterranean diet enjoins patients with diabetes to consume diets characterized by high amounts of olive oil, nuts, fruits, cereals, legumes, whole grains and vegetables; moderate amounts of fish, poultry and dairy products; and low amounts of red meat, sweets, highly processed foods and refined grains [43,44].

The findings of the present study revealed that the participants planned their meals by eating healthy sources of carbohydrates, fats and protein, eating the recommended quantity of foods, and engaging in healthy eating habits. These findings are in tandem with the findings of other studies conducted among patients with diabetes in Tanzania and Ethiopia [45,46]. The findings of the current study suggest that the participants generally have adequate knowledge and skills on meal planning due to their healthy habits and practices. Nonetheless, it is worth noting that the participants do not often consume high amounts of olive oil and nuts. It is therefore recommended that patients with diabetes in Ghana should be educated and sensitized to consume high amounts of olive oil and nuts as part of the healthy sources of carbohydrates, fats and protein that they take to ensure optimal health.

## Conclusions

Optimal nutritional management of diabetes mellitus by patients living with the condition involves their ability to choose and consume healthy sources of carbohydrates, protein and fats; eat the right quantity of foods; possess adequate skills to efficiently plan meals; and read and use food labels. This study showed that the participants ate the right quantity of foods; consumed healthy carbohydrates, proteins and fats; had adequate meal planning skills; and had deficient knowledge and skills in the reading and usage of food labels. Hence, the participants of this study generally engaged in optimal nutritional management of their condition due to their healthy dietary habits, practices and preferences. Patients with diabetes may thus be able to successfully engage in optimal nutritional management of their condition when they have the right abilities and skills to engage in healthy eating, and are given adequate education on healthy eating, including education on proper reading and usage of food labels by health care professionals.

## Supporting information

**S1 File. Interview guide.**
(DOCX)

## Acknowledgments

The authors are grateful to the participants of this study for their participation.

## Author Contributions

**Conceptualization:** Kwabena Opoku-Addai, Kwadwo Ameyaw Korsah, Gwendolyn Patience Mensah.

**Data curation:** Kwabena Opoku-Addai, Gwendolyn Patience Mensah.

**Formal analysis:** Kwabena Opoku-Addai, Kwadwo Ameyaw Korsah, Gwendolyn Patience Mensah.

**Investigation:** Kwabena Opoku-Addai.

**Methodology:** Kwabena Opoku-Addai, Kwadwo Ameyaw Korsah, Gwendolyn Patience Mensah.

**Supervision:** Kwadwo Ameyaw Korsah, Gwendolyn Patience Mensah.

**Writing – original draft:** Kwabena Opoku-Addai.

**Writing – review & editing:** Kwabena Opoku-Addai, Kwadwo Ameyaw Korsah, Gwendolyn Patience Mensah.

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
