## [Decision Letter · Decision Letter 0]

15 Nov 2021

PONE-D-21-20315Nutritional self-care practices and skills of patients with diabetes mellitus: A study at a tertiary hospital in GhanaPLOS ONE

Dear Dr. Gwendolyn Patience Mensah,

Thank you for submitting your manuscript to PLOS ONE. After careful consideration, we feel that it has merit but does not fully meet PLOS ONE’s publication criteria as it currently stands. Therefore, we invite you to submit a revised version of the manuscript that addresses the points raised during the review process. Please submit your revised manuscript by Dec 30 2021 11:59PM. If you will need more time than this to complete your revisions, please reply to this message or contact the journal office at plosone@plos.org. Please include the following items when submitting your revised manuscript:A rebuttal letter that responds to each point raised by the academic editor and reviewer(s). You should upload this letter as a separate file labeled 'Response to Reviewers'.A marked-up copy of your manuscript that highlights changes made to the original version. You should upload this as a separate file labeled 'Revised Manuscript with Track Changes'.An unmarked version of your revised paper without tracked changes. You should upload this as a separate file labeled 'Manuscript'.

We look forward to receiving your revised manuscript.

Kind regards,

Sheikh Mohd Saleem, MBBS, MD

Academic Editor

PLOS ONE

Journal Requirements:

● A clean copy of the edited manuscript (uploaded as the new *manuscript* file).

3. When reporting the results of qualitative research, we suggest consulting the COREQ guidelines: http://intqhc.oxfordjournals.org/content/19/6/349. In this case, please consider including more information on the number of interviewers, their training and characteristics; and please provide the interview guide used.

Additional Editor Comments:

Thankyou Very much for submission to PLOS ONE. we have received the comments from our reviewers and based on that we have decided to have a major revision for you paper. Kindly address each point as suggested by the reviewer's while revising your manuscript. I would suggest to use an academic writer or professional English writing tool for handling your manuscript. Best wishes

Reviewers' comments:

Reviewer's Responses to Questions

**Comments to the Author**

1. Is the manuscript technically sound, and do the data support the conclusions?

Reviewer #1: Yes

Reviewer #2: No

2. Has the statistical analysis been performed appropriately and rigorously? 

Reviewer #1: N/A

Reviewer #2: Yes

3. Have the authors made all data underlying the findings in their manuscript fully available?

Reviewer #1: Yes

Reviewer #2: No

4. Is the manuscript presented in an intelligible fashion and written in standard English?

Reviewer #1: No

Reviewer #2: No

5. Review Comments to the Author

Reviewer #1: Most of the key words not MeSH terms. Scientific or otherwise basis of chosing 15 participants not explained. Results section of abstract does not show any numerical or percentage value. Literature review on subject is not up to the mark. Sampling technique is not mentioned. Values of findings in results should be mentioned as percentage points or qualifiers like half of participants, one third, one fourth etc. Absolute numbers are avoided in qualitative studies. Grammer need to be improved.

Reviewer #2: The present study has many limitations, majorly sample size is very less and the due to this it cant provide definite conclusions. inclusion authors considered both type 1 and type 2 diabetes.

Authors may present the the nutritional status and in the tabular form.

6. PLOS authors have the option to publish the peer review history of their article (what does this mean?). If published, this will include your full peer review and any attached files.

Reviewer #1: No

Reviewer #2: No

---

## [Author Response · Author response to Decision Letter 0]

29 Dec 2021

Response to Review Decision 29/12/2021

Manuscript number: PONE-D-21-20315

Title of the manuscript: Nutritional self-care practices and skills of patients with diabetes mellitus: A study at a tertiary hospital in Ghana

Dear Editor, 

We are so grateful for giving us the opportunity to revise our manuscript for resubmission. The suggestions from the reviewers and editor have greatly improved the manuscript. All the concerns raised by the reviewers have been addressed accordingly. We thank you and the reviewers for your comments. 

Below are our responses to the points raised by you and the reviewers. 

All authors have agreed to the revisions made in the manuscript for submission. 

Yours sincerely, a

Gwendolyn Patience Mensah (PhD) 

(corresponding author) 

EDITOR: 

1. We suggest you thoroughly copyedit your manuscript for language usage, spelling, and grammar. 

The manuscript has been edited by an academic writer who does professional scientific editing. All the grammatical errors have been corrected by the academic writer who edited the manuscript. 

She is Ms. Vicki Igglesden, Nelson Mandela University, Port Elizabeth, South Africa. 

2. When reporting the results of qualitative research, we suggest consulting the COREQ guidelines: http://intqhc.oxfordjournals.org/content/19/6/349. In this case, please consider including more information on the number of interviewers, their training and characteristics; and please provide the interview guide used.

On lines 10 to 12 of page 6, information on who conducted the interviews has been clearly stated. The interview guide for the study has also been provided as an attachment to the manuscript. 

REVIEWER #1: 

1. Is the manuscript presented in an intelligible fashion and written in standard English? PLOS ONE does not copyedit accepted manuscripts, so the language in submitted articles must be clear, correct, and unambiguous. Any typographical or grammatical errors should be corrected at revision, so please note any specific errors here.

Reviewer #1: No

The manuscript has been edited by an academic writer who does professional scientific editing. The typographical and grammatical errors have been corrected by the academic writer who edited the manuscript. 

2. Most of the key words are not MeSH terms. 

The keywords that were not MeSH terms have now been changed to MeSH terms. This can be found on the last line of page 2. 

3. Scientific or otherwise basis of choosing 15 participants not explained. 

The scientific rationale or basis for choosing 15 participants has now been clearly stated on lines 15 to 18 of page 5. 

4. Results section of abstract does not show any numerical or percentage value. 

The results section of the abstract has now been edited to include numerical values. This can be found on page 2. 

5. Literature review on subject is not up to the mark. 

Literature has been reviewed on the subject. On lines 15 to 18 of page 3 and lines 7 to 11 of page 4, new literature has been added to the manuscript. 

6. Sampling technique is not mentioned. 

The sampling technique that was used for the study has been clearly stated on line 11 of page 5. 

7. Values of findings in results should be mentioned as percentage points or qualifiers like half of participants, one third, one fourth etc. Absolute numbers are avoided in qualitative studies. 

This has been corrected on all the pages of the results section of the manuscript. 

8. Grammar needs to be improved.

The manuscript has been edited by an academic writer who does professional scientific editing. All the grammatical errors have been corrected by the academic writer who edited the manuscript. 

REVIEWER #2: 

1. Is the manuscript technically sound, and do the data support the conclusions? 

Reviewer #2: No 

The manuscript has been edited by an academic writer who does professional scientific editing. All the grammatical and technical errors have been corrected by the academic writer who edited the manuscript. The scientific basis for the sample size used has been stated on lines 15 to 18 of page 5. We believe the corrections effected have made the manuscript technically sound. 

2. Have the authors made all data underlying the findings in their manuscript fully available? The PLOS Data policy requires authors to make all data underlying the findings described in their manuscript fully available without restriction, with rare exception (please refer to the Data Availability Statement in the manuscript PDF file). The data should be provided as part of the manuscript or its supporting information, or deposited to a public repository. For example, in addition to summary statistics, the data points behind means, medians and variance measures should be available. If there are restrictions on publicly sharing data—e.g. participant privacy or use of data from a third party—those must be specified.

Reviewer #2: No 

We believe that the data underlying the findings of the study has been indicated or provided in the manuscript. The results section of the manuscript contains numerous direct quotes from the participants of the study. 

3. Is the manuscript presented in an intelligible fashion and written in standard English?  PLOS ONE does not copyedit accepted manuscripts, so the language in submitted articles must be clear, correct, and unambiguous. Any typographical or grammatical errors should be corrected at revision, so please note any specific errors here.

Reviewer #2: No 

The manuscript has been edited by an academic writer who does professional scientific editing. The typographical and grammatical errors have all been corrected by the academic writer who edited the manuscript. 

4. The present study has many limitations, majorly sample size is very less and due to this it can’t provide definite conclusions.

The scientific rationale for the sample size used has been indicated on lines 15 to 18 of page 5. The sample size was determined by saturation. We believe this justifies the conclusions drawn. 

5. Inclusion: authors considered both type 1 and type 2 diabetes. 

This has clearly been indicated on lines 12 and 13 of page 5.

---

## [Decision Letter · Decision Letter 1]

7 Mar 2022

Nutritional self-care practices and skills of patients with diabetes mellitus: A study at a tertiary hospital in Ghana

PONE-D-21-20315R1

Dear Dr. Mensah,

We’re pleased to inform you that your manuscript has been judged scientifically suitable for publication and will be formally accepted for publication once it meets all outstanding technical requirements.

Kind regards,

Sheikh Mohd Saleem, MBBS, MD

Academic Editor

PLOS ONE

Additional Editor Comments (optional):

Reviewers' comments:

Reviewer's Responses to Questions

**Comments to the Author**

1. If the authors have adequately addressed your comments raised in a previous round of review and you feel that this manuscript is now acceptable for publication, you may indicate that here to bypass the “Comments to the Author” section, enter your conflict of interest statement in the “Confidential to Editor” section, and submit your "Accept" recommendation.

Reviewer #1: All comments have been addressed

Reviewer #2: All comments have been addressed

2. Is the manuscript technically sound, and do the data support the conclusions?

Reviewer #1: Yes

Reviewer #2: Partly

3. Has the statistical analysis been performed appropriately and rigorously? 

Reviewer #1: Yes

Reviewer #2: Yes

4. Have the authors made all data underlying the findings in their manuscript fully available?

Reviewer #1: Yes

Reviewer #2: Yes

5. Is the manuscript presented in an intelligible fashion and written in standard English?

Reviewer #1: Yes

Reviewer #2: Yes

6. Review Comments to the Author

Reviewer #1: (No Response)

Reviewer #2: No other comments to the authors, authors answered all my queries. Hence the the present manuscript format can be accepted.

7. PLOS authors have the option to publish the peer review history of their article (what does this mean?). If published, this will include your full peer review and any attached files.

Reviewer #1: **Yes: **PK Anand

Reviewer #2: **Yes: **Ramu Adela

---

## [Editor Report · Acceptance letter]

15 Mar 2022

PONE-D-21-20315R1 

Nutritional self-care practices and skills of patients with diabetes mellitus: A study at a tertiary hospital in Ghana. 

Dear Dr. Mensah:

I'm pleased to inform you that your manuscript has been deemed suitable for publication in PLOS ONE. Congratulations! Your manuscript is now with our production department. 

Kind regards, 

on behalf of

Dr. Sheikh Mohd Saleem 

Academic Editor

PLOS ONE